# TKRM: A Formal Knowledge Representation Method for Typhoon Events

**Peng Ye [1,2], Xueying Zhang [1,2,*], Ge Shi [1,2], Shuhui Chen [1,2], Zhiwen Huang [1,2] and Wei Tang [3]**

[1]  Key Laboratory of Virtual Geographic Environment, Ministry of Education, Nanjing Normal University, Nanjing 210023, China; 161301027@stu.njnu.edu.cn (P.Y.); 161301020@stu.njnu.edu.cn (G.S.); 191302096@stu.njnu.edu.cn (S.C.); 181302068@stu.njnu.edu.cn (Z.H.)

[2]  Jiangsu Center for Collaborative Innovation in Geographical Information Resource Development and Application, Nanjing 210023, China

[3]  CMA Public Meteorological Service Center, Beijing 100081, China; now_tang@163.com

*   Correspondence: zhangsnowy@163.com; Tel.: +86-025-8589-1561

**Abstract:** Typhoon events can cause serious environmental damage and economic losses. Understanding the development of typhoon events will provide valuable knowledge for disaster prevention and mitigation. In the age of big data, the sharp contrast between the sudden increase of mass information and the lack of a knowledge appreciation mechanism appears. There is an urgent need to promote the transformation of information services to knowledge services in the field of hazard management. Knowledge representation, as a strategy for symbolizing and formalizing knowledge, affects knowledge acquisition, storage, management, and application, and is the basis and prerequisite for the implementation of knowledge services. Based on the evolution law of typhoon events and human cognitive habits, a formal knowledge representation method for typhoon events (TKRM) is proposed in this paper. First, by analyzing the evolution characteristics of typhoon events, the TKRM framework with three layers consisting of "event–process–state" was constructed, which was used to describe the knowledge composition and relationship of the different granularity of typhoon events. Second, the formal representation of the TKRM framework was formed by using a finite state machine (FSM) as a reference, taking time and location as the basic conditions, and extending the hierarchical and parallel representation mechanism. Finally, the rationality and practical value of the TKRM were verified using a case study.

**Keywords:** typhoon event; typhoon evolution process; knowledge representation model; knowledge formalization; process knowledge modeling

## 1. Introduction

Typhoons are natural hazards occurring in the northwest Pacific and South China Sea, which are accompanied by intense releases of enormous amounts of energy [1,2]. When a typhoon passes through an area, it will not only cause strong winds, rainstorms, and huge waves, but also easily lead to flooding, debris flow, landslides, and other secondary disasters [3]. Every year, typhoon events cause direct economic losses amounting to over $100 billion to over 10 million people, making it one of the most destructive hazards in the world [4].

The reason why a typhoon is so destructive is not only the bad weather, but also the complex and changeable evolution process that brings great uncertainty to hazard management. For a long time, people have accumulated a lot of information about typhoons, such as historical information and real-time observation information. Due to the lack of effective information mining and sorting,

it is difficult to generate typhoon knowledge using only information technology. Knowledge is the understanding of the links between repeatable information that is formed through the processing, interpretation, selection, and transformation of information [5]. The generation of a typhoon hazard is often driven by the evolution of the typhoon such that the evolution law and pattern of a typhoon for a given time and location is a kind of valuable knowledge. There is an urgent requirement for an objective description of the evolution process, the key nodes and impacts of a typhoon, and condensing the related concepts and concept relations of typhoon hazards to provide knowledge support for hazard prevention and reduction in terms of a strong scope, sufficient demonstration, and strict logic.

Knowledge representation is the basis and premise of knowledge application. On the one hand, knowledge needs to be represented before it can be organized and stored. On the other hand, knowledge representation determines the subsequent processing efficiency and application scope. Knowledge representation is a description or a group of conventions of knowledge, which is the modeling, symbolization, and formalization of knowledge. In the age of big data, knowledge representation is closely related to the strategy of transforming human knowledge into a system control structure that can be processed by computers [6]. Knowledge representation requires not only the encoding of knowledge in the form of language, mathematics, or formal logic [7,8], but more importantly, the construction of a cognitive representation model [9]. At present, the modeling of typhoon events can be divided into two types: the static conceptual model and the timeslice snapshot model. For the static conceptual model, it is usually based on an analysis of related basic concepts in the field of the typhoon, deriving the hierarchical relationship using the semantic classification of the concepts, and forming a tree structure [10–12]. This kind of model can clearly show the different characteristics of the typhoon, but it also ignores the dynamic characteristics of the typhoon. As for the timeslice snapshot model, the contents of different indicators describing typhoons are formed into a time series according to the sequence of their generation time [13,14]. Although this expression reflects the dynamic changes of the typhoon to some extent, it usually only describes part of the characteristics of the typhoon (such as typhoon intensity, typhoon track, etc.), and cannot express the characteristics of the typhoon in different locations at the same time.

The essence of typhoon hazard management is managing the changing situation of a typhoon in time and determining the status and evolution trend of a typhoon hazard. However, both the static conceptual model and the timeslice snapshot model lack a description of the typhoon state or typhoon process, and do not fully express all kinds of knowledge needed for hazard management. It is noteworthy that through the study of cognitive science, linguistics, computer informatics, and other disciplines, human beings memorize and understand the real-world primarily in terms of "events," which are important components of knowledge [15,16]. Thus, the event-centered knowledge representation model emerged [17–19]. Since an event involves many kinds of entities, it is a knowledge unit with larger granularity than an entity. It can express more advanced and complicated semantic information. The use of an event as the core of knowledge modeling better reflects the characteristics of the objective world and emphasizes the dynamics of the event itself [20–23]. Models of event ontologies, event Petri networks, event frameworks, etc. have been generated [24–26] and are being applied in the areas of emergencies, public opinion events, social security events, and so on [27–29].

Compared with a generalized event representation, there are several factors that require more consideration in a typhoon event: (1) A typhoon is a geographical phenomenon, therefore time and location are the basic conditions for its existence. In the existing event models, time, location, and other attributes are juxtaposed, which do not reflect the characteristics of time and location as the basic framework [30,31]. (2) Describing the dynamics of events by depicting changes in states or intervals is still equivalent to recording a snapshot of an event at different time segments. Due to the lack of a mechanism for continuous gradual expression in the model, the continuous evolution of typhoon events cannot be effectively expressed and organized. (3) People's cognition of events varies according to their needs and purposes. Therefore, it is necessary to abstract typhoon events and describe them using different granularities to fully understand the external causes and internal connections.

To construct the knowledge hierarchy of typhoon evolution, this study proposed a formal knowledge representation method for typhoon events (TKRM). Based on the analysis of the content characteristics of typhoon events, a knowledge representation model was constructed to describe the basic components and their relationships of knowledge and form structured knowledge of typhoon events. Furthermore, a formal method for typhoon event knowledge was established to realize the formalization of typhoon event knowledge. The main innovation of this study is reflected in the following two aspects:

(1) Combined with the evolution law of typhoon events and human cognitive habits, this study summarized and abstracted the content and characteristics of typhoon events, constructed the framework of the TKRM, and unified the basic components of knowledge of typhoon events and their relationships.

(2) Based on the structural characteristics of typhoon event knowledge, the typical finite state machine method was extended to implement the formal representation of typhoon event knowledge.

The arrangement of the main sections of this paper is as follows: Section 2 presents the methodology for the TKRM, Section 3 provides the experiment and evaluation of the TKRM, Section 4 discusses the reasonableness and application value of the TKRM, and Section 5 provides conclusions and prospective future work.

## 2. Methodology

The proposed TKRM model adopts the method of hierarchical modeling and constructs a typhoon event knowledge representation model using the three layers of "event–process–state". In the TKRM model, the multi-dimensional feature of "event-state" is extracted, and the "state-process" is associated with describing the continuity of events, which provides more abundant semantic information and a more complete dynamic expression. Based on this, the formal representation of the TKRM model was realized by extending the finite state machine (FSM). The specific technical flow of the TKRM is shown in Figure 1.

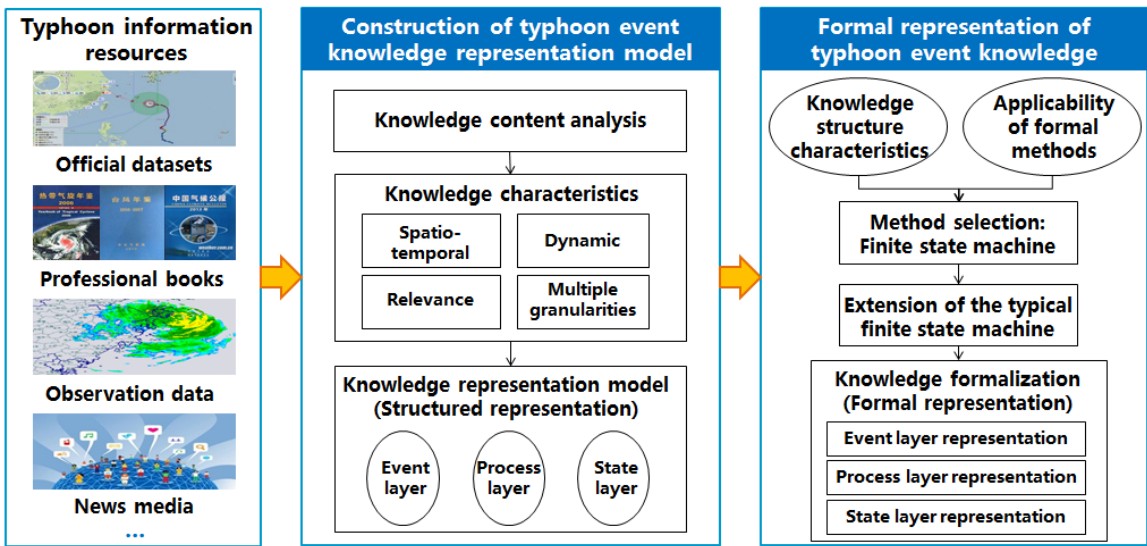

**Figure 1.** Technical flowchart of the typhoon event knowledge representation method (TKRM).

### 2.1. TKRM Framework

Typhoon events are the products of combined action of a pregnant environment (E), causation factor (C), and sustaining body (S) [32]. A typhoon event includes not only the development and evolution of the typhoon itself (C), but also the effect of the surface environment (E) on the development

of the typhoon, as well as the series of impacts of the typhoon on the human and the human living environment (S). In particular, if a typhoon event has a destructive effect on human production and life during the life cycle, due to the vulnerability of the sustaining body in society, a typhoon hazard will evolve into a disaster. A typhoon event has a destructive evolution process, which mainly includes four stages: (1) Formation—the typhoon increases gradually from the initial depression circulations to a cyclone where the wind scale reaches level 8. (2) Development—the cyclone continues to develop until the central air pressure reaches its lowest value and the wind speed reaches its maximum. (3) Continuance—the wind speed is no longer increased and the air pressure is no longer reduced, but the influence range is gradually expanded. (4) Decline—the central wind speed of the cyclone decreases and the air pressure increases or it enters the middle latitude area and becomes an extratropical cyclone. Furthermore, the evolution of a typhoon event also exhibits the following characteristics: (1) Spatiotemporal—the spatiotemporal consistency of typhoon development cannot be neglected and the separation of space and time is impossible. (2) Dynamic—the intensity of the causing factors, the range of influence, and the degree of the hazard are constantly changing. (3) Relevance—typhoons are affected by the external environment but also have a destructive effect on other things. (4) Multiple granularities—due to different actual needs, people have a different understanding of the time and space scope and degree of typhoon events.

Because typhoon events go through a process from occurring to ending, people's cognition of typhoon events progress from parts to the whole, from one-sidedness to all-sidedness. In the knowledge modeling of typhoon events, it is necessary to follow the content characteristics of typhoon events and human cognitive habits to achieve a comprehensive understanding of typhoon events.

### 2.1.1. Frame Structure

To address the multiple granularities of typhoon events, the proposed typhoon representation model should also include multi-granularity. Granularity is the basic unit of knowledge, and the size of the granularity is a measure of the level of knowledge abstraction and the knowledge included. According to the requirements of problem-solving, knowledge is divided into different angles or levels, and each knowledge block is a level of knowledge granularity [33,34]. For typhoon events, people need both integrated knowledge with a larger granularity, such as the start and end time of the typhoon, the range of impact, and the overall hazard level, as well as detailed knowledge with smaller granularity, such as the landing location of the typhoon, a typhoon's wind scale at a given moment, etc. Usually, people begin by understanding the different states in an event, each of which is a slice in the sequence of events. Many states are connected in series according to logic, and the process situation and all kinds of relations of the event development are presented. Therefore, events with a larger granularity can be divided into processes and states with a smaller granularity, and states with a smaller granularity can be aggregated into processes and events with a larger granularity [35]. Based on the hierarchical modeling, this paper presents a knowledge representation model with the three layers of "event–process–state" for typhoon events (Figure 2).

### 2.1.2. Event Layer

In essence, people's cognitions of typhoon events abstract the complex typhoon weather system and its natural environment. If the occurrence of a typhoon causes damage to human production and life, it will further cover the abstraction of the social environment. Because of the multi-granularity of typhoon events, the spatiotemporal span of typhoon events is variable. Typhoon events with different granularities have a hierarchical structure, and events with a larger spatiotemporal span cover multiple events with smaller spatiotemporal span. When facing specific application scenarios, people's cognitions of typhoon events are always within a certain spatiotemporal span such that typhoon events with a certain granularity have a fixed spatiotemporal span. Moreover, the event layer in the model is not only the determination of the cognitive granularity, but also the basic definition and expression of

related concepts of the whole event, such as the typhoon name, typhoon number, hazard grade, and so on.

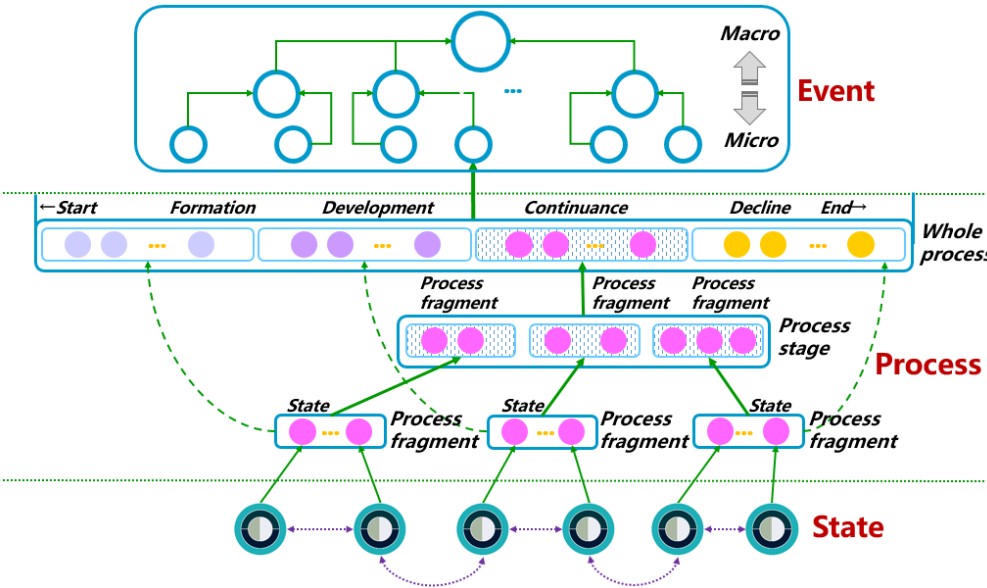

**Figure 2.** Frame structure of the typhoon event knowledge representation model.

### 2.1.3. State Layer

The state is the situation of typhoon events in a certain section, which is characterized by a series of characteristics [36]. The state of events exists in time and at a location, and time and location are the basic frameworks for describing the state. The formation of hazards is the result of the effect of causation factors on sustaining bodies under the influence of a pregnant environment. A typhoon, as a causation factor, has attributes and behavioral characteristics. At the same time, the generation, development, and change of typhoons are all affected by the environment. In response, a typhoon causes great damage to the surrounding environment. Therefore, this paper proposes to characterize the state of typhoon events using five dimensions: time, location, attribute, behavior, and influence (Figure 3).

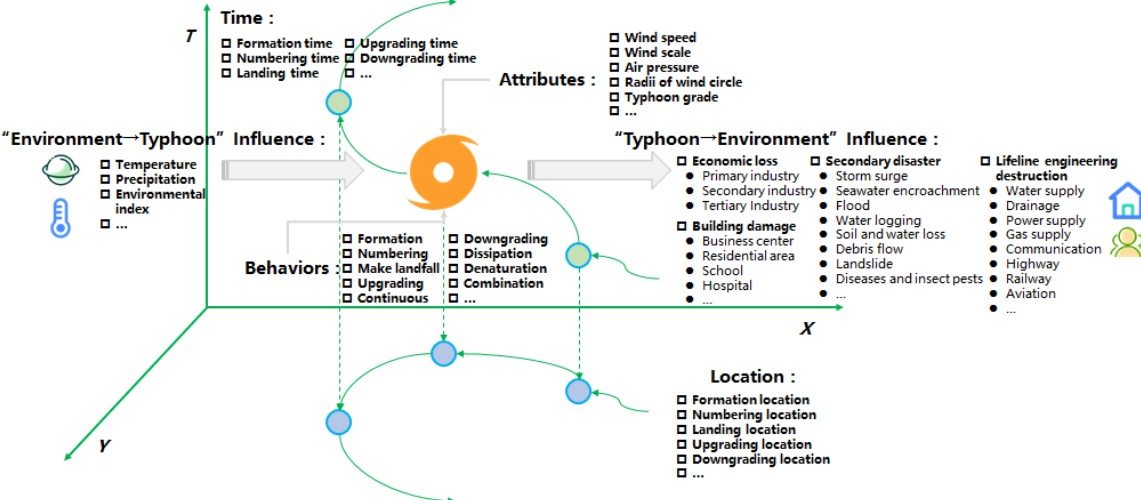

**Figure 3.** Different characteristics of representational states.

(1) Time and location as the basic framework: Time and location describe the spatiotemporal feature of the state, which is the premise of the existence of other dimensional features. Time is a measure of the order in which states occur and location records the space or spatial coordinates where states occur.

(2) Attribute and behavior as core characteristics: Attribute and behavior describe the typhoon's characteristics, which is an important reflection of the typhoon's development degree. Among them, the relatively static attribute feature is used to depict the appearance of the typhoon, and the dynamic behavior feature is used to depict the action of the typhoon.

(3) Influence as a peripheral interaction feature: Influence describes the interaction between a typhoon and its surroundings. Influence is an indispensable part of determining the extent of the hazard that can describe the typhoon hazard more comprehensively.

When the characteristics of state change, the type of state will also change. In particular, as a causation factor, a typhoon is the most critical factor causing a hazard and determining the extent of the hazard; therefore, the attribute and behavior are the basis for determining the type of state. According to the development process of typhoon events and relevant standard specifications, the main state types of typhoon events are classified (Table 1). For example, Typhoon Lekima made landfall in Wenling City, Zhejiang Province, with a wind scale at level 16, and its state type was "super typhoon + landfall".

**Table 1.** Types of states included in typhoon events.

| Reference Feature | Classification | Classification Basis | Description |
|---|---|---|---|
| Typhoon attributes [37] | Weaker than tropical depression | There is no clear circulation center, with the maximum sustained wind speed below 10.8 m/s and wind scale below level 6 | Tropical disturbance |
| | Tropical depression | Maximum sustained wind speed of 10.8–17.1 m/s and wind scale of level 6–7 | Tropical cyclone |
| | Tropical storm | Maximum sustained wind speed of 17.2–24.4 m/s and wind scale of level 8–9 | |
| | Severe tropical storm | Maximum sustained wind speed of 24.5–32.6 m/s and wind scale of level 10–11 | |
| | Typhoon | Maximum sustained wind speed of 32.7–41.4 m/s and wind scale of level 12–13 | |
| | Severe typhoon | Maximum sustained wind speed of 41.5–50.9 m/s and wind scale of level 14–15 | |
| | Super typhoon | Maximum sustained wind speed above 51.0 m/s and wind scale above level 16 | |
| | Extratropical cyclone | The cyclone appears in the mid-high latitudes, the pressure of the cyclone center is lower than the surroundings, and has the characteristics of a cold center | |
| Typhoon behaviors | Formation | Formation of tropical cyclones in the ocean | Tropical cyclone reaches tropical depression |
| | Numbering | Number and name in order of occurrence | Tropical cyclone reaches tropical storm |
| | Landfall | Typhoon center moves from ocean to land | |
| | Upgrading | Typhoon wind scale rises | |
| | Continuance | Typhoon wind scale remains unchanged | |
| | Downgrading | Typhoon wind scale drops | |
| | Dissipation | Typhoon wind scale continue to decline, and there is no obvious circulation center | Dissipation, denaturation, and combination are all signs of typhoon cessation, and only one of them will appear in the typhoon life cycle. |
| | Denaturation | Tropical cyclone turns into an extratropical cyclone | |
| | Combination | One or more typhoons are combined into one typhoon | |

### 2.1.4. Process Layer

The process is the development of the event, which can describe the trend of the typhoon event. The interrelation of different states is the basis of the process, and the process can express the continuous gradual change of characteristics and mechanisms between the states. Thus, the process includes not only a certain number of states but also the relationships between different states (Figure 4).

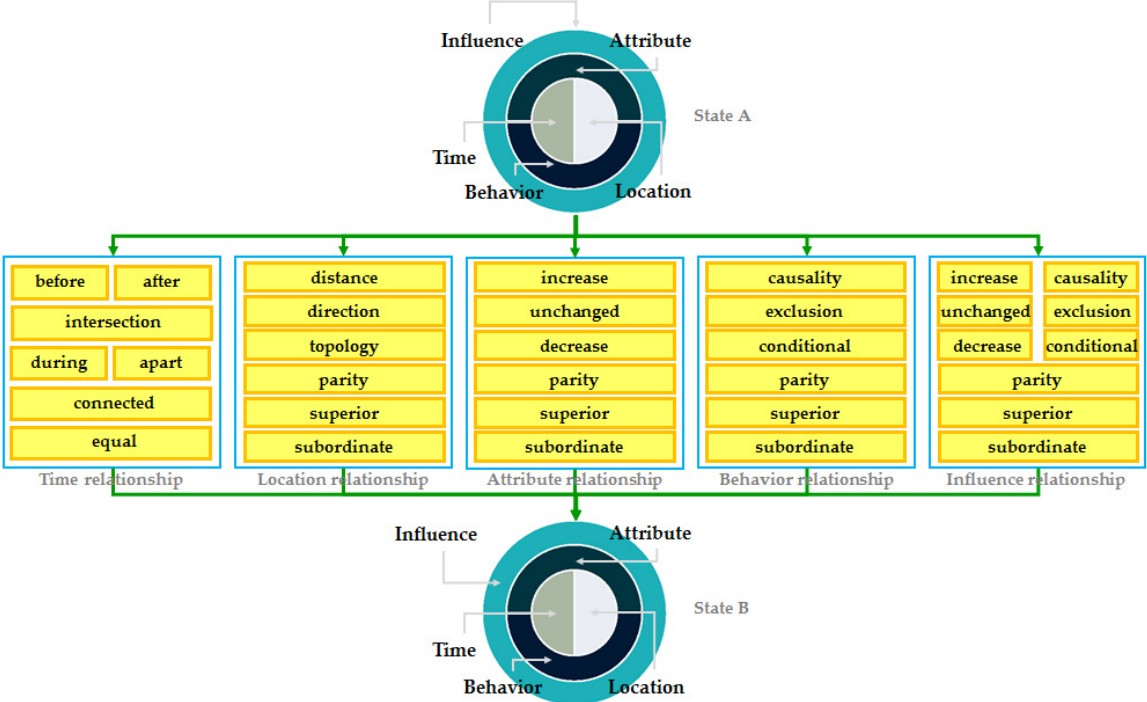

**Figure 4.** Relationship types between states.

Because of the different states of the constituent processes, granularity differences also exist between processes. According to the spatiotemporal span that the process can cover, the process is divided into the whole process, process stage, and process fragment [38] (Figure 5). It is not only easy to express the multi-scale of the process by abstracting the process hierarchically, but it is also more conducive to reflecting the evolution law of typhoon events. The whole process can describe the beginning and end of an event. Process stages are divided according to the coarse granularity of the typhoon life cycle, including the four stages of formation, development, continuance, and decline. Process fragments are the finer-grained partition within a process stage. The larger granularity process implicitly reflects the evolution mechanism of events to realize the continuous gradual expression of the smaller granularity process. The smaller granularity process carries the larger granularity process and records more detailed events.

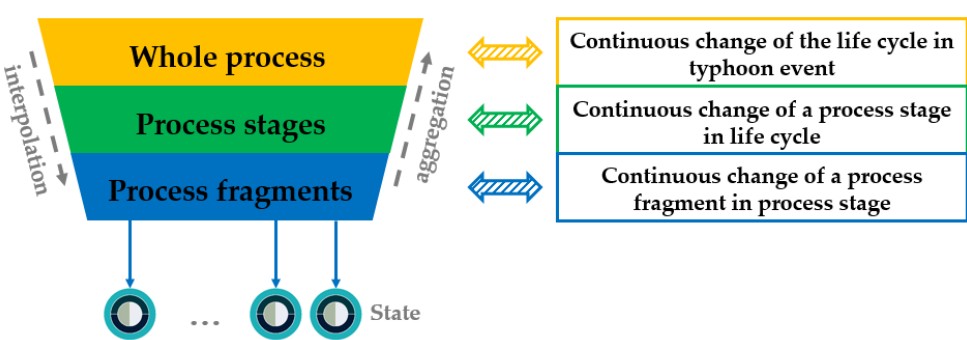

**Figure 5.** Hierarchical structure of the process.

*2.2. TKRM Formalization*

2.2.1. Typical Finite State Machine

Formalization is an indispensable part of knowledge representation to create an unambiguous and computable knowledge representation. The finite state machine (FSM) is a classical formalization

method that is mainly used to describe the sequence of states that an object experiences in its life cycle and how different states respond to various behaviors from the outside world. The state of an object in an FSM is finite, and the state transition occurs due to triggering conditions [39]. An FSM can be expressed as a five-tuple:

$$M = (Q, \sum, \delta, q_0, F),\qquad(1)$$

where $Q = \{q_0, q_1, \ldots, q_n\}$ is a set of finite states. At any given time, an FSM can only be in a determinate state $q_i$. $\sum = \{\sigma_1, \sigma_2, \ldots, \sigma_n\}$ is a finite set of input characters, and an FSM can accept only one definite input $\sigma_j$ at any given time. $\delta: q_i \times \sigma_j \rightarrow q'$ is a set of state transfer rules. When the FSM is in a certain state $q_i \in Q$, $\sigma_j \in \sum$ is accepted as an input character and it can be transferred to the next state $q'$ through $\delta$. $q_0$ is the initial state where $q_0 \in Q$. F is the set of finalized states where $F \subseteq Q$. $q_0$ and F can be omitted as an alternative [40]. In general, for the operating mechanism of an FSM, a state $q_m$ with a given input $\sigma_j$ will be transferred to a new state $q_n$, which is determined by $\delta$.

Previous studies used decision tree and association rule methods to examine tropical cyclones [41, 42], but the application scenarios of these methods have different focuses. An association rule method is mainly used to discover the correlation among different elements in an event [43], and a decision tree is a method is used to obtain decision results through probability analysis [44,45]. However, in this study, an FSM was chosen instead of decision tree and association rule methods mainly because FSM is better at reflecting and representing the dynamics of events. Each element of the typhoon event is organized according to spatiotemporal clues in the TKRM model, while an FSM is suitable for describing sequential and logical objects. In comparison, an FSM fits better with the representation requirements of the TKRM model.

However, there are still some shortcomings in the direct application of FSM to the representation of the TKRM framework: (1) There is a lack of independent representation of the characteristics of time and location in an FSM. (2) An FSM is not suitable for describing hierarchical characteristics, whereas the TKRM framework presents a nested hierarchy. Moreover, unlike a simple state in a hierarchical state machine (HSM), changes at one layer in the TKRM do not migrate to either the upper or lower layers. (3) An FSM is weak at describing concurrent typhoons, which is concerning given the phenomenon of "binary typhoons" or even "triple typhoons" exists in typhoon hazards.

2.2.2. Extension of a Finite State Machine

The typical FSM was extended based on the formal requirements of the TKRM (Figure 6). This is mainly reflected in the following three aspects:

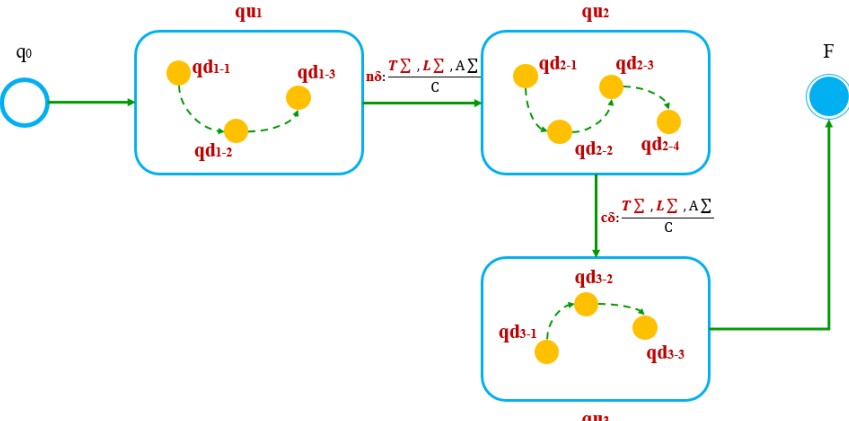

**Figure 6.** Extended finite state machine (FSM) structure and state transfer diagram.

(1) The time and location are modeled separately, and a description mechanism for the time and location characteristics was added. The extended FSM is represented as follows:

$$M = (Q, T\sum, L\sum, A\sum, \delta, q_0, F) \tag{2}$$

where $T\sum = \{t\sigma_0, t\sigma_1, ..., t\sigma_n\}$ is a set of time conditions, $L\sum = \{l\sigma_0, l\sigma_1, ..., l\sigma_n\}$ is a set of location conditions, $A\sum = \{a\sigma_0, a\sigma_1, ..., a\sigma_n\}$ is a set of input conditions other than time and location, and is also defined as the set of trigger conditions. Regarding the $\delta$, $\delta$: $q_i \times (t\sigma_j \cup l\sigma_j \cup a\sigma_j) \to q'$, the description of the transfer rules of time and location were also added, which was also convenient for the description and expression of the time and location relationships.

(2) Based on Statecharts structure [46,47], the FSM was extended to a hierarchical form:

$$(M_1, M_2, \ldots, M_n, R), n \geq 3 \tag{3}$$

where on the one hand, the states in the upper state machine are subdivided according to the spatiotemporal granularity, and the states in the lower state machine can be decomposed. In Figure 6, $Qu = \{qu_1, qu_2, qu_3\}$ is the state of the upper state machine, and $Qd_1 = \{qd_{1\text{-}1}, qd_{1\text{-}2}, qd_{1\text{-}3}\}$ is the state of the lower state machine; then, $Qd_1 \subseteq qu_1$, R for mapping rules at different layers are explained. On the other hand, the state transfer does not change the hierarchy to which the state belongs. For $\delta d_{1\text{-}2}$: $qd_{1\text{-}1} \times (t\sigma d_{1\text{-}1} \cup l\sigma d_{1\text{-}1} \cup a\sigma d_{1\text{-}1}) \to qd_{1\text{-}2}$, then $qd_{1\text{-}1}, qd_{1\text{-}2} \in Qd_1$.

(3) To express the typhoon event in parallel, the FSM is further expanded to:

$$M = (Q, T\sum, L\sum, A\sum, C, \delta, q_0, F) \tag{4}$$

where $C = \{c_0, c_1, \ldots, c_n\}$ is the key feature set of each state in Q, and the start and end time of the state needs to be recorded. $\delta$ is differentiated into a conventional transfer $c\delta$ and a parallel transfer $p\delta$. When $p\delta$ is executed, it enters the next state and does not exit the previous state. Each $\delta$ can match the feature set C. The range D of each feature item in $\delta$ can be described as $c\delta_i = D(c_{i0}) \cup D(c_{i1}) \cup \ldots \cup D(c_{in})$, $t\sigma \not\subset c_{(i-1)t} \cap t\sigma \subseteq c_{it}$; $p\delta_j = D(c_{j0}) \cup D(c_{j1}) \cup \ldots \cup D(c_{jn})$, $t\sigma \subseteq c_{(j-1)t} \cap t\sigma \subseteq c_{jt}$. The state transfer can only occur if all the characteristic change requirements are met. When the time condition $t\sigma$ does not belong to the time characteristic $c_{(i-1)t}$ of the previous state, a conventional transfer is performed; otherwise, a parallel transfer is performed. $\delta$ can be described using a two-dimensional table (Table 2).

**Table 2.** Schematic set of state transfer rules.

| States | $D(c_{i0})$ | ... | $D(c_{it})$ | $D(c_{in})$ | $t\sigma \subseteq c_{(i\text{-}1)t}? \cap t\sigma \subseteq c_{it}?$ | Type |
|---|---|---|---|---|---|---|
| $q_0 \to q_1$ | [a, b] | ... | (m, n] | (r, s) | False ∩ True | $c\delta$ |
| $q_1 \to q_2$ | [c, d] | ... | (p, q] | (u, v) | True ∩ True | $p\delta$ |

According to Table 2, if the input conditions $T\Sigma$, $L\Sigma$, and $A\Sigma$ satisfy the feature terms $a\sigma_{10} \in [a, b]$, $\ldots$, $a\sigma_{1n} \in (r, s)$, and $t\sigma_1 \not\subset c_{0t} \cap t\sigma_1 \subseteq c_{1t}$, then the state $q_0$ is transferred to $q_1$ through $c\delta$. If $a\sigma_{10} \in [c, d]$, $\ldots$, $a\sigma_{1n} \in (u, v)$, $t\sigma_1 \subseteq c_{1t} \cap t\sigma_1 \subseteq c_{2t}$, then the state $q_1$ is transferred to $q_2$ through $c\sigma$.

2.2.3. Formal Representation of the TKRM

Using the extended finite state machine method above, the TKRM is formally represented.

**Definition 1.** *The state machine model of TKRM is:*

$$\text{TKRM} = (E, P, S, R) \tag{5}$$

*TKRM is composed of an event-layer state machine E, a process-layer state machine P, and a state-layer state machine S. R defines the state mapping relationship between the layers.*

**Definition 2.** *The structure of the event-layer state machine E is:*

$$E = (Q_e, T\textstyle\sum_e, L\textstyle\sum_e, A\textstyle\sum_e, C_e, \delta_e) \tag{6}$$

**Definition 3.** *The structure of the process-layer state machine P is:*

$$P = (Q_p, T\textstyle\sum_p, L\textstyle\sum_p, A\textstyle\sum_p, C_p, \delta_p, R_p) \tag{7}$$

*Additionally, P can be further stratified into $P_1, P_2, \ldots, P_n$ with different spatiotemporal granularities according to $(M_1, M_2, \ldots, M_n, R)$ rules.*

**Definition 4.** *The structure of the state-layer state machine S is:*

$$S = (Q_s, T\textstyle\sum_s, L\textstyle\sum_s, A\textstyle\sum_s, C_s, \delta_s, F_s, R_s) \tag{8}$$

*The formulas for the parameters in the state machine E, P, and S are shown in Table 3.*

In state machine E, $Q_e$ is the set of typhoon events. $T\sum_e$ is the set of the times of events and $L\sum_e$ is the set of the locations of events. $A\sum_e$ is the set of trigger conditions of all event transfer. $C_e$ is the set of key features of typhoon events. $\delta_e$ is the set of event transfer rules, including conventional transfer and parallel transfer rules.

In state machine P, $Q_p$ is the set of typhoon processes. $T\sum_p$ is the set of the times of processes and $L\sum_p$ is the set of the locations of processes. $A\sum_p$ is the set of trigger conditions of all process transfers. $C_e$ is the set of key features of the typhoon processes. $\delta_p$ is the set of process transfer rules, excluding parallel transfer. Each event $eq_i$ in E can be subdivided into multiple Ps, and each process $pq_i$ in P can be further subdivided into multiple smaller-granularity Ps. $R_p$ describes the mapping rules between different layers of state machines. When the state machine P ends, the state $eq_i$ in the E state machine to which it belongs also transfers to the next state $eq_{(i+1)}$.

In state machine S, $Q_s$ is the set of typhoon states. $T\sum_s$ is the set of the times of states and $L\sum_s$ is the set of the locations of states. $A\sum_s$ is the set of trigger conditions of all state transfers. $C_s$ is the set of key features of typhoon states. $\delta_s$ is the set of state transfer rules, excluding parallel transfer. F is the set of finalized states of typhoon, $F_s \subseteq Q_s$. Each process $pq_i$ in P can be subdivided into multiple Ss. $R_s$ describes the mapping rules between different layers of state machines. When the state machine S enters the final state F, the state $pq_i$ in the P state machine to which it belongs also needs to transfer to the next state $pq_{(i+1)}$.

**Table 3.** Formulas for parameters in the state machine model of TKRM.

| | TKRM = (E, P, S, R) | | |
|---|---|---|---|
| | **E = ($Q_e$, $T\sum_e$, $L\sum_e$, $A\sum_e$, $C_e$, $\delta_e$)** | **P = ($Q_p$, $T\sum_p$, $L\sum_p$, $A\sum_p$, $C_p$, $\delta_p$, $R_p$)** | **S = ($Q_s$, $T\sum_s$, $L\sum_s$, $A\sum_s$, $C_s$, $\delta_s$, $F_s$, $R_s$)** |
| Q | $Q_e$ = {$eq_i$ = [$eq_{i0}$|$eq_{i1}$]typhoon event, $eq_{i0}$ = typhoon number, $eq_{i1}$ = typhoon name|$eq_i \in Q_e$, i = 0, 1, … , n} | $Q_p$ = {$pq_i$ = [$pq_{i0}$|$pq_{i1}$]typhoon process, $pq_{i0}$ = process number, $pq_{i1}$ = process name|$pq_i \in Q_p$, i = 0, 1, … , n} | $Q_s$ = {$sq_0$ = formation, $sq_1$ = numbering, $sq_2$ = tropical depression + upgrade, $sq_2$ = tropical storm + upgrade, … ,$sq_j$ = super typhoon + landfall, … , $sq_n$ = combination} |
| $T\sum$ | $T\sum_e$ = {$t_e\sigma_i$|$t_e\sigma_i \in T\sum_e$, i = 0, 1, … , n} | $T\sum_p$ = {$t_p\sigma_i$|$t_p\sigma_i \in T\sum_p$, i = 0, 1, … , n} | $T\sum_s$ = {$t_s\sigma_i$|$t_s\sigma_i \in T\sum_s$, i = 0, 1, … , n} |
| $L\sum$ | $L\sum_e$ = {$l_e\sigma_i$|$l_e\sigma_i \in L\sum_e$, i = 0, 1, … , n} | $L\sum_p$ = {$l_p\sigma_i$|$l_p\sigma_i \in L\sum_p$, i = 0, 1, … , n} | $L\sum_s$ = {$l_s\sigma_i$|$l_s\sigma_i \in L\sum_s$, i = 0, 1, … , n} |
| $A\sum$ | $A\sum_e$ = {$a_e\sigma_i$ = [$a_e\sigma_{i0}$|$a_e\sigma_{i1}$]typhoon formation, $a_e\sigma_{i0}$ = typhoon number, $a_e\sigma_{i1}$ = typhoon name|$a_e\sigma_i \in A\sum_e$, i = 0, 1, … , n} | $A\sum_p$ = {$a_p\sigma_i$ = enter[$a_p\sigma_{i0}$|$a_p\sigma_{i1}$]process, $a_p\sigma_{i0}$ = process number, $a_p\sigma_{i1}$ = process name|$a_p\sigma_i \in A\sum_p$, i = 0, 1, … , n} | $A\sum_s$ = {$a_s\sigma_0$ = formation, $a_s\sigma_1$ = numbering, $a_s\sigma_2$ = landfall, $a_s\sigma_3$ = upgrading, $a_s\sigma_4$ = continuance, $a_s\sigma_5$ = downgrading, $a_s\sigma_6$ = dissipation, $a_s\sigma_7$ = denaturation, $a_s\sigma_8$ = combination} |
| C | $C_e$ = {$ec_i$|$ec_i \in C_e$, i = 0, 1, … , n}, $ec_i$ = {$ec_{i0}$ = typhoon number, $ec_{i1}$ = typhoon name, $ec_{it}$ = [start time, end time], $ec_{il}$ = {related locations}, $ec_{i2}$ = maximum typhoon grade, $ec_{i3}$ = hazard situation} | $C_p$ = {$pc_i$|$pc_i \in C_p$, i = 0, 1, … , n}, $pc_i$ = {$pc_{i0}$ = process number, $pc_{i1}$ = process name, $pc_{it}$ = [start time, end time], $pc_{il}$ = [start location, end location], $pc_{i2}$ = [start level, end level], $pc_{i5}$ = typhoon moving direction, $pc_{i6}$ = typhoon moving speed, $pc_{i7}$ = wind strength change rate, $pc_{i8}$ = hazard impact change degree} | $C_s$ = {$sc_i$|$sc_i \in C_s$, i = 0, 1, … , n}, $sc_i$ = {$sc_{i0}$ = state number, $sc_{i1}$ = state name, $sc_{it}$ = time, $sc_{il}$ = location, $sc_{i2-1}$ = attribute-wind speed, $sc_{i2-2}$ = attribute-wind scale, … , $sc_{i3}$ = behavior, …, $sc_{i4-1}$ = influece-casualties, $sc_{i4-2}$ = impact-economic loss, …} |
| $\delta$ | $eq_{(i+1)}$ = $\delta_{ei}$($eq_i$, $t_e\sigma_i$, $l_e\sigma_i$, $a_e\sigma_i$) <br> $c\delta_e$ = {$n_et_i$|$n_et_i \in \sigma_e$, i = 0, 1, … , n}, $n_et_i$ = D($ec_{it}$) ∪ D($ec_{il}$), $t_e\sigma_i \not\subset ec_{(i-1)2}$ ∩ $t_e\sigma_i \subseteq ec_{i2}$} <br> $p\delta_e$ = {$c_et_i$|$c_et_i \in \sigma_e$, i = 0, 1, … , n}, $c_et_i$ = D($ec_{it}$) ∪ D($ec_{il}$), $t_e\sigma_i \subseteq ec_{(i-1)2}$ ∩ $t_e\sigma_i \subseteq ec_{i2}$ | $pq_{(i+1)}$ = $\delta_{pi}$($pq_i$, $tp\sigma_i$, $lp\sigma_i$, $ap\sigma_i$), <br> $c\delta_p$ = {$npt_i$|$npt_i \in \sigma_p$, i = 0, 1, … , n}, $npt_i$ = D($pc_{i0}$) ∪ D($pc_{i1}$) ∪ … ∪ D($pc_{i8}$), $tp\sigma_i \not\subset pc_{(i-1)t}$ ∩ $tp\sigma_i \subseteq pc_{it}$ | $sq_{(i+1)}$ = $\delta_{si}$($sq_i$, $ts\sigma_i$, $ls\sigma_i$, $as\sigma_i$), <br> $c\delta_s$ = {$nst_i$|$nst_i \in \sigma_s$, i = 0, 1, … , n}, $nst_i$ = D($sc_{i0}$) ∪ D($sc_{i1}$) ∪ … ∪ D($sc_{ij}$), $ts\sigma_i \not\subset sc_{(i-1)t}$ ∩ $ts\sigma_i \subseteq sc_{it}$ |
| F | - | - | $F_s$ = {$sf_0$ = dissipation, $sf_1$ = denaturation, $sf_2$ = combination} |

## 3. Experiment and Results

Super Typhoon Lekima (International Code: 1909) is the ninth named tropical storm of the 2019 Pacific typhoon season. "Lekima" was named by the Japan Meteorological Agency at 15:00 on 4 August 2019, and was upgraded to a typhoon by the China Central Meteorological Station at 5:00 on 7 August. It made landfall on the coast of Chengnan Town, Wenling City, Zhejiang Province, at 01:45 on 10 August. Then, the typhoon crossed Zhejiang and Jiangsu provinces and moved into the Yellow Sea. At 20:50 on 11 August, it made landfall again on the coast of Huangdao District, Qingdao City, Shandong Province. Since then, the typhoon moved into the Bohai Sea and continued to weaken. Eventually, the typhoon stopped numbering at 14:00 on 13 August [48]. Based on the TKRM, the knowledge of the typhoon Lekima event was modeled and formalized.

For the event-layer state machine E, the event set $Q_e$ was defined as: {..., $eq_{1908}$ = 1908 typhoon Francisco event, $eq_{1909}$ = 1909 typhoon Lekima event, $eq_{1910}$ = 1910 typhoon Krosa event, $eq_{1911}$ = 1911 typhoon Bailu event, ...}. The set of key features of typhoon events $C_e$ is defined in Table 4.

**Table 4.** Examples of key features of typhoon events $C_e$.

| $ec_0$ | $ec_t$ | $ec_l$ | $ec_1$ | $ec_2$ | ... |
|---|---|---|---|---|---|
| ... | ... | ... | ... | ... | ... |
| 1908 | (19-08-02,19-08-06) | (152.80°/20.10°, ... , 129.40°/35.20°) | Francisco | wind scale level 14 | ... |
| 1909 | (19-08-04, 19-08-13) | (131.90°/17.40°, ... , 119.90°/37.50°) | Lekima | wind scale level 17 and above | ... |
| 1910 | (19-08-06, 19-08-16) | (142.80°/18.40°, ... , 136.70°/41.30°) | Krosa | wind scale level 14 | ... |
| 1911 | (19-08-21, 19-08-26) | (132.20°/15.70°, ... , 113.20°/24.60°) | Bailu | wind scale level 11 | ... |
| ... | ... | ... | ... | ... | ... |

According to the definition in $C_e$, the event transfer rules $\delta_e$ were modeled (Table 5).

**Table 5.** Examples of typhoon event transfer rules.

| States | $D(ec_{it})$ | $D(ec_{il})$ | Result |
|---|---|---|---|
| ... | ... | ... | ... |
| $eq_{1908} \rightarrow eq_{1909}$ | (19-08-04, 19-08-13) | (131.90°/17.40°, ... , 119.90°/37.50°) | parallel transfer |
| $eq_{1909} \rightarrow eq_{1910}$ | (19-08-06, 19-08-16) | (142.80°/18.40°, ... , 136.70°/41.30°) | parallel transfer |
| $eq_{1910} \rightarrow eq_{1911}$ | (19-08-21, 19-08-26) | (132.20°/15.70°, ... , 113.20°/24.60°) | conventional transfer |
| ... | ... | ... | ... |

When E received the trigger conditions $t_e\sigma_{1909}$ = 19-08-04, $l_e\sigma_{1909}$ = 131.90°/17.40°, they were judged by the combination of $ec_{1908}$ and $ec_{1909}$, and transfer rule $\delta_{e1909}$. Due to $t_e\sigma_{1909} \subseteq ec_{1908t} \cap t_e\sigma_{1909} \subseteq ec_{1909t}$, the event parallel transfer to "1909 typhoon Lekima event" and "1908 typhoon Francisco event" continued.

For the process-layer state machine P, only the process stages were represented to simplify the model. The process set $Q_p$ was defined as: {$pq_0$ = tropical disturbance (initial state), $pq_1$ = formation stage, $pq_2$ = development stage, $pq_3$ = continuance stage, $pq_4$ = decline stage}. The set of key features of the typhoon processes $C_p$ is defined as Table 6.

**Table 6.** Examples of key features of typhoon processes $C_p$.

| $pc_0$ | $pc_t$ | $pc_l$ | $pc_1$ | $pc_2$ | $pc_3$ | $pc_4$ | ... |
|---|---|---|---|---|---|---|---|
| $P_1$ | (19-08-04 14: 00, 19-08-05 02:00) | (131.90°/17.40°, ... , 130.50°/17.90°) | formation stage | (level 8, level 9) | north-west | 4 km/h | ... |
| $P_2$ | (19-08-05 02: 00, 19-08-08 21:00) | (130.50°/17.90°, ... , 124.90°/24.40°) | development stage | (level 9, level 17 and above) | north-west | 22 km/h | ... |
| $P_3$ | (19-08-08 21: 00, 19-08-10 03:00) | (124.90°/24.40°, ... , 121.20°/28.40°) | continuance stage | (level 17 and above, level 15) | north-northwest | 15 km/h | ... |
| $P_4$ | (19-08-10 03: 00, 19-08-13 14:00) | (121.20°/28.40°, ... , 120.30°/37.90°) | decline stage | (level 15, level 15) | north-northeast | 12 km/h | ... |

According to the definition in $C_p$, the process transfer rules $\delta_p$ were modeled (Table 7).

**Table 7.** Examples of process transfer rules for typhoon Lekima event.

| States | $D(pc_{it})$ | $D(pc_{i1})$ | $D(pc_{i2})$ | ... | Result |
|---|---|---|---|---|---|
| $pq_0 \rightarrow pq_1$ | (19-08-04 14: 00, 19-08-05 02:00) | (131.90°/17.40°, ..., 130.50°/17.90°) | (level 8, level 9) | ... | conventional transfer |
| $pq_1 \rightarrow pq_2$ | (19-08-05 02: 00, 19-08-08 21:00) | (130.50°/17.90°, ..., 124.90°/24.40°) | (level 9, level 17 and above) | ... | conventional transfer |
| $pq_2 \rightarrow pq_3$ | (19-08-08 21: 00, 19-08-10 03:00) | (124.90°/24.40°, ..., 121.20°/28.40°) | (level 17 and above, level 15) | ... | conventional transfer |
| $pq_3 \rightarrow pq_4$ | (19-08-10 03: 00, 19-08-13 14:00) | (121.20°/28.40°, ..., 120.30°/37.90°) | (level 15, level 7) | ... | conventional transfer |

When P received the trigger conditions $t_p\sigma_2$ = 19-08-05 02: 00, $l_p\sigma_2$ = 130.50°/17.90°, and $a_p\sigma_{22}$ = level 9, they were judged by the combination of $pc_1$ and $pc_2$, and transfer rule $\delta_{p2}$. Due to $t_p\sigma_2 \not\subset c_{1t} \cap t_p\sigma_2 \subseteq pc_{2t}$, the process conventional transfer to "development stage" and "formation stage" was finished.

Since all of the above were modeling the processes of the "1909 typhoon Lekima event," $pq_0$–$pq_4$ were in the lower state machine of $eq_{1909}$ in E; as such, $R_p = \{Q_p \subseteq eq_{1909}, T\sum_p \subseteq t_e\sigma_{1909}, L\sum_p \subseteq l_e\sigma_{1909}\}$.

For the state-layer state machine S, only part of the typhoon states were represented to simplify the model. The status set $Q_s$ was defined as: $\{sq_0$ = formation, $sq_a$ = numbering, $sq_b$ = typhoon + upgrading, $sq_c$ = super typhoon + continuance, $sq_d$ = super typhoon + landfall, $sq_e$ = tropical storm + landfall, $sq_f$ = tropical storm + downgrading, $sq_g$ = dissipation, ...$\}$. The set of key features of typhoon states $C_s$ is defined as Table 8.

**Table 8.** Examples of key features of typhoon states $C_s$.

| $sc_0$ | $sc_t$ | $sc_1$ | $sc_1$ | $sc_{2-1}$ | $sc_{2-2}$ | ... | $sc_3$ | ... | $sc_{4-1}$ | ... |
|---|---|---|---|---|---|---|---|---|---|---|
| $s_0$ | 19-08-04 14:00 | 131.90°/17.40° | formation | 18 m/s | level 8 | ... | formation | ... | - | ... |
| $s_a$ | 19-08-04 15:00 | 131.50°/17.10° | numbering | 18 m/s | level 8 | ... | numbering | ... | - | ... |
| $s_b$ | 19-08-07 05:00 | 128.10°/19.80° | typhoon + upgrading | 33 m/s | level 12 | ... | upgrading | ... | - | ... |
| $s_c$ | 19-08-08 21:00 | 124.90°/24.40° | super typhoon + continuance | 62 m/s | level 17 and above | ... | continuance | ... | - | ... |
| $s_d$ | 19-08-10 01:45 | 121.40°/28.30° | super typhoon + landfall | 52 m/s | level 16 | ... | landfall | ... | - | ... |
| $s_e$ | 19-08-11 20:50 | 120.10°/36.10° | tropical storm + landfall | 23 m/s | level 9 | ... | landfall | ... | - | ... |
| $s_f$ | 19-08-12 17:00 | 119.30°/37.20° | tropical storm + downgrading | 20 m/s | level 8 | ... | downgrading | ... | - | ... |
| $s_g$ | 19-08-13 14:00 | 119.90°/37.50° | dissipation | 16 m/s | level 7 | ... | dissipation | ... | 14.024 million people were affected and 57 died | ... |

According to the definition in $C_s$, the state transfer rules $\delta_s$ were modeled (Table 9).

**Table 9.** Examples of state transfer rules for the typhoon Lekima event.

| States | $D(sc_{it})$ | $D(sc_{i1})$ | $D(sc_{i2-1})$ | $D(sc_{i2-2})$ | ... | $D(sc_{i3})$ | ... | Result |
|---|---|---|---|---|---|---|---|---|
| $sq_0$ | 19-08-04 14:00 | 131.90°/17.40° | 18 m/s | level 8 | ... | formation | ... | conventional transfer |
| $sq_{(a-1)} \rightarrow sq_a$ | 19-08-04 15:00 | 131.50°/17.10° | 18 m/s | level 8 | ... | numbering | ... | conventional transfer |
| $sq_{(b-1)} \rightarrow sq_b$ | 19-08-07 05:00 | 128.10°/19.80° | 33 m/s | level 12 | ... | upgrading | ... | conventional transfer |
| $sq_{(c-1)} \rightarrow sq_c$ | 19-08-08 21:00 | 124.90°/24.40° | 62 m/s | level 17 and above | ... | continuance | ... | conventional transfer |
| $sq_{(d-1)} \rightarrow sq_d$ | 19-08-10 01:45 | 121.40°/28.30° | 52 m/s | level 16 | ... | landfall | ... | conventional transfer |
| $sq_{(e-1)} \rightarrow sq_e$ | 19-08-11 20:50 | 120.10°/36.10° | 23 m/s | level 9 | ... | landfall | ... | conventional transfer |
| $sq_{(f-1)} \rightarrow sq_f$ | 19-08-12 17:00 | 119.30°/37.20° | 20 m/s | level 8 | ... | downgrading | ... | conventional transfer |
| $sq_{(g-1)} \rightarrow sq_g$ | 19-08-13 14: 00 | 119.90°/37.50° | 16 m/s | level 7 | ... | dissipation | ... | conventional transfer |
| ... | ... | ... | ... | ... | ... | ... | ... | ... |

When S received the trigger conditions $t_s\sigma_d$ = 19-08-10 01:45, $l_s\sigma_d$ = 121.40°/28.30°, $a_s\sigma_{d2-2}$ = level 16, and $a_s\sigma_{d3}$ = landfall, it was judged by the combination of $sc_d$ and transfer rule $\delta_{sd}$. The state conventional transfer to "super typhoon + landfall" was undertaken.

Since all of the above were modeling the states of the "1909 typhoon Lekima event," $sq_0$–$sq_g$ were in the lower state machine of $pq_1$–$pq_4$ in P; as such, $R_{e1} = \{sq_0 \subseteq pq_1, t_s\sigma_0 \subseteq t_p\sigma_1, l_s\sigma_0 \subseteq l_p\sigma_1\}$, $R_{e2} = \{(sq_a, sq_b) \subseteq pq_2, (t_s\sigma_a, t_s\sigma_b) \subseteq t_p\sigma_2, (l_s\sigma_a, l_s\sigma_b) \subseteq l_p\sigma_2\}$, $R_{e3} = \{(sq_c, sq_d) \subseteq pq_3, (t_s\sigma_c, t_s\sigma_d) \subseteq l_p\sigma_3, (l_s\sigma_c, l_s\sigma_d) \subseteq l_p\sigma_3\}$, $R_{e4} = \{(sq_e, sq_f, sq_g) \subseteq pq_4, (t_s\sigma_e, t_s\sigma_f, t_s\sigma_g) \subseteq t_p\sigma_4, (l_s\sigma_e, l_s\sigma_f, l_s\sigma_g) \subseteq l_p\sigma_4\}$.

For comparison, the static conceptual model and timeslice snapshot model were also used to represent the typhoon Lekima event, and the structure of different models is shown in Figure 7. The

case study shows that the static conceptual model focused on the basic concepts and classification system of typhoon field knowledge, but could not reflect the dynamics of the typhoon. The timeslice snapshot model focused on the changes of the typhoon itself, and lacked information about the pregnant environment, sustaining body, and other important hazard contents. More importantly, the typhoon information in different locations at the same time could not be organized. The TKRM is a dynamic expression of typhoon events oriented to a "multi-state, whole process," which conforms to the evolution law of events. In this state, various characteristics of a typhoon at a certain spatiotemporal node are described. The state itself does not emphasize ordering according to time, location, or other characteristics. The states are interconnected to form a process according to different relationships such that the process mechanism for describing the continuous gradual change is possible. With the addition of states, the process is expressed in more detail. Moreover, because of the different relationship types between states, it can describe the changing processes of typhoon events in different dimensions. Additionally, the formal representation of a typhoon event is consistent and complete, and has a strong scalability. Therefore, the TKRM is feasible for practical applications.

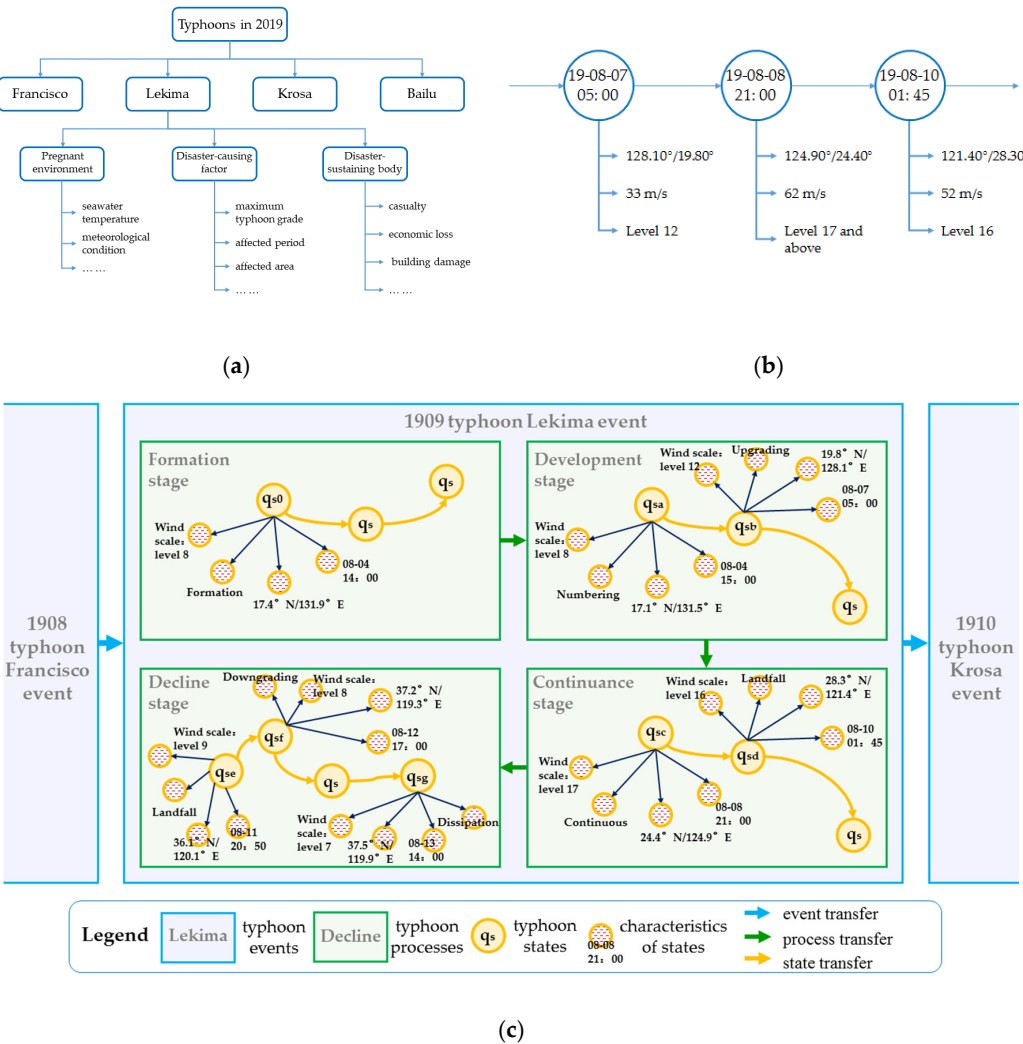

**Figure 7.** The typhoon Lekima event examples with structures of the static conceptual model, timeslice snapshot model, and TKRM model: (**a**) typhoon Lekima event structure based on the static conceptual model, (**b**) typhoon Lekima event structure based on the timeslice snapshot model, and (**c**) typhoon Lekima event structure based on the TKRM model.

## 4. Discussion

Typhoon events are usually the result of the interactions within an atmosphere, surface environment, and human society system, in which there are a lot of uncertain object relations and huge concept systems. At present, there is a lack of a unified knowledge description mechanisms for typhoon events, which leads to the difficulty in integrating typhoon knowledge with an appreciation mechanism using a large amount of complex information [49–51]. Because of the urgent demand for information services to transfer to knowledge services in the hazard management field, it is crucial to construct a reasonable and effective typhoon knowledge representation method.

### 4.1. The Advantage of the TKRM Compared with Conventional Typhoon Models

As demonstrated in Figure 7, the proposed model TKRM better described the characteristics of the typhoon Lekima event than the static conceptual model and timeslice snapshot model in the following two aspects:

1.  The conventional modeling method takes the typhoon as a solid object and abstracts the static concept of the typhoon to form a classification structure. Particularly the static conceptual model ignores the detailed process and internal motivation of the spatiotemporal variation of a typhoon, and the expression of all kinds of non-classification relationships is complex.
2.  Compared with other types of events, typhoon events have more significant spatiotemporal variability. On the one hand, at two different time nodes, the location of a typhoon center and its attributes characteristics will change; on the other hand, for two different locations, the period and degree of typhoon impact are completely different. For example, when a typhoon moves to location A, location A is most strongly affected by the typhoon. However, for location B, which is far from location A, it is only affected by the outer wind ring. The timeslice snapshot model takes a typhoon to be a series of time sequence changes to reflect the change of a typhoon in the time dimension. However, the typhoon itself has a wide range of influence, and there are significant changes in spatial dimensions. The timeslice snapshot model cannot reflect the spatiotemporal variability of typhoon events.

Through comparison and analysis, the representation model of the TKRM takes time and location as the frame and extracts the process and state of events using a multi-granular classification, which is more suitable for the representation of typhoon events.

### 4.2. Application Potential of the TKRM

The TKRM includes two parts: the representation model and the formal method. The TKRM model gives descriptions with different granularities from a complete stereoscopic view that is a direct expression of the existence of typhoon events. The formal method of the TKRM provides the possibility for the operability and computation of typhoon event knowledge, and it adapts to the demand for knowledge services in a big data environment. Therefore, TKRM is not only the basis of general knowledge services, such as retrieval, analysis, and prediction, but can also provide targeted support for typhoon prevention and reduction.

The key to managing typhoon events lies in the timely judgment and accurate prediction of a typhoon trend; thus, it is difficult to analyze the typhoon evolution completely and clearly with a single knowledge service. Typhoon event knowledge based on the TKRM can provide a global reference for understanding the evolution of typhoon events, mainly including the following:

1.  Spatiotemporal pattern: By employing spatiotemporal analysis, many spatiotemporal characteristics of typhoon events can be clearly defined, such as the distribution of landing locations, the duration times, the regional differences of risks, the regional distribution of hazard grades, etc. [52,53]. During the frequent periods of typhoon events, relevant high-risk areas need to make emergency plans as soon as possible.

2. Evolution law: The evolution logic of spatiotemporal, sequential, and causal relationships exists between different typhoon states [54–56]. By analyzing and combining these relationships, it reveals the general trend of typhoon events, such as the evolution of the typhoon track and the evolution of the typhoon intensity.

3. Activity mode: By combining the cognition of the spatiotemporal pattern and evolution law, the main types of event development and change can be summarized [57]. For example, analyzing the spatiotemporal characteristics of the trajectory of the typhoon track, the typhoon essentially moves along one of three trajectory modes: "west line," "west to north," and "north line" [58].

4. Internal mechanism: Based on a comprehensive consideration of the spatiotemporal pattern, evolution law, and activity mode, the internal changing features of each element in a typhoon event, as well as the rules and principles of the interaction and interconnection of elements in a certain spatiotemporal environment, are obtained. It provides a real and reliable reference for typhoon prediction, early warning, and emergency decision-making [59,60].

*4.3. Recommendations for Future Research*

The knowledge abstraction and representation of typhoon events is only the initial stage of knowledge engineering construction. To realize the extension of a typhoon knowledge service in the context of big data, there are still many scientific problems to be solved, such as knowledge acquisition, management, and application. In the future, we need to study the method of building a knowledge base based on the TKRM. This included not only the shallow semantic understanding of typhoon events from multi-source data, but also the more advanced knowledge acquisition methods, such as event relation extraction, event process recognition, and knowledge reasoning should be established. Additionally, the combination of the TKRM-based knowledge service model with hazard prevention and mitigation, emergency decision-making, and other fields will be explored to achieve the effect of supporting the knowledge guarantee for early crisis warning, monitoring, command, evaluation, and other decision-making work. The combination method mainly includes the following:

1. Early warning: The evolution law of similar historical typhoon events should be fully utilized [61]. Comparing the characteristics of the current typhoon event with other typhoon events in history, the hazard severity and evolution trend of similar typhoon events can provide a reference for the current typhoon event prediction and early warning.

2. Hazard monitoring: The TKRM clearly defines the content scope of typhoon events. It should focus on acquiring and perceiving the content in terms of the time, location, attributes, behavior, and influence.

3. Prevention command: Judging the state of a typhoon event is the key to an effective emergency response. To this end, the TKRM can guide and support decision-makers to accurately grasp the current typhoon situation [62,63]. For example, when the typhoon is in the "super typhoon + landfall" state, the affected areas need to start emergency plans, and relevant departments begin to implement hazard rescue. When the typhoon is in a "tropical storm + downgrading" state, the relevant departments need to carry out hazard relief and a series of rehabilitation and reconstruction work.

4. Hazard assessment: Based on the abundant content covered in the TKRM, it can meet the demands of various assessments, such as casualties, economic losses, and secondary disasters [64].

## 5. Conclusions

An event is the basic cognitive unit of a human's cognitive world. With the development of typhoon research, the requirement of knowledge modeling for typhoon events is increasing. Based on the characteristic analysis of the typhoon event process, a knowledge representation model for typhoon event evolution was established. The TKRM provides abundant spatiotemporal semantics and a more complete dynamic process for describing typhoon events through the hierarchical representation

mechanism of "multi-state, whole process." Based on the framework, the FSM was extended to implement the formal description of TKRM and the typhoon events were expressed in terms of its spatiotemporal, dynamic, correlation, and granularity properties. Compared with the conventional models, the proposed TKRM highlights the characteristics of time and location as the basic framework, which meet the needs of expressing the spatiotemporal variability of typhoon events, and by abstracting and describing typhoon events using different granularities, it can provide comprehensive support for the acquisition, management, and application of typhoon event knowledge.

**Author Contributions:** Conceptualization, P.Y. and X.Z.; methodology, P.Y. and X.Z.; validation, P.Y. and G.S.; formal analysis, P.Y. and S.C.; investigation, X.Z.; data curation, P.Y., Z.H., and W.T.; writing—original draft preparation, P.Y.; writing—review and editing, P.Y., X.Z., and G.S.; visualization, P.Y.; project administration, X.Z. All authors have read and agreed to the published version of the manuscript.

**Funding:** This research was funded by the National Natural Science Foundation of China (grant nos. 41631177, 41971337, and 41671393).

**Acknowledgments:** The authors thank Jing Liu for her critical reviews and constructive comments.

**Conflicts of Interest:** The authors declare no conflict of interest.

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
