# Peer review of "TKRM: A Formal Knowledge Representation Method for Typhoon Events"

_sustainability, doi:10.3390/su12052030_

Round 1

Reviewer 1 Report

Comments on “TKRM: A Formal Knowledge Representation Method for Typhoon Events”

Comments:

To integrate and accumulate typhoon knowledge effectively and provide decision support for disaster management, the authors have proposed a formal knowledge representation method for 16 typhoon events (TKRM). The topic seems interesting. Why should we build such a method to represent typhoon events in the first place? Meteorologists can access necessary best track data for tropical cyclones freely. To be more specifically, what can we do with this knowledge representation of typhoons? Can this knowledge-based system help understand and forecast typhoons? Can this system improve the efficiency of forecasters? It might be helpful to better organize the historical information of typhoons. I would suggest the authors to better discuss the significance and contribution of this work.  

Lines 13-16: Is this a decision support system for typhoons? Can we use this system for typhoon analysis and prediction or disaster management (Leung et al., 2012; Ding et al. 2015)?

Line 18: consists of

Line 50: “are composed into” might not be the best option in this context.

Line 289: What is the trigger condition? How can we use this trigger condition?

Lines 300-302: In this method similar to “decision tree” or “association rule”? For example, Yang et al. (2011) and Zhang et al. (2013) applied decision tree and association rule methods to examine tropical cyclones. Some conditions could lead to a decision on the state of tropical cyclones.

Line 348: There is lack of

Lines 394-397: The authors wrote “In addition, the combination of the TKRM-based knowledge service model with disaster prevention and mitigation, emergency decision-making and other fields will be explored to achieve the effect of supporting the knowledge guarantee for crisis early warning, monitoring, command, evaluation and other decision-making work.” Can the authors use some examples to demonstrate this aspect?

Line 409: The authors use “granularity” many times, but never giving an explanation. This should be properly defined for the readers to understand its meaning.

I can spot “Error! Reference source not found” throughout the manuscript. Please make sure this is corrected.

Reference:

Ding Y. et al.  (2015) An integrated geospatial information service system for disaster management in China. International Journal of Digital Earth 8:11, pages 918-945.

Zhang, W., Y. Leung, and J.C. Chan, 2013: The Analysis of Tropical Cyclone Tracks in the Western North Pacific through Data Mining. Part I: Tropical Cyclone Recurvature. J. Appl. Meteor. Climatol., 52, 1394–1416

Yang, R., J. Tang, and D. Sun, 2011: Association Rule Data Mining Applications for Atlantic Tropical Cyclone Intensity Changes. Wea. Forecasting, 26, 337–353,

Leung Y. et al. (2012) A novel web-based system for tropical cyclone analysis and prediction, International Journal of Geographical Information Science, 26:1, 75-97

Author Response

Point 1: To integrate and accumulate typhoon knowledge effectively and provide decision support for disaster management, the authors have proposed a formal knowledge representation method for 16 typhoon events (TKRM). The topic seems interesting. Why should we build such a method to represent typhoon events in the first place? Meteorologists can access necessary best track data for tropical cyclones freely. To be more specifically, what can we do with this knowledge representation of typhoons? Can this knowledge-based system help understand and forecast typhoons? Can this system improve the efficiency of forecasters? It might be helpful to better organize the historical information of typhoons. I would suggest the authors to better discuss the significance and contribution of this work.

Response 1: We agree with the reviewer's comment that adds content about the significance and contribution of the study. At the age of big data, the phenomenon of "mass of data, information explosion, and hard to find knowledge" is common. It also leads to the accumulation of typhoon-related information resources in the long-term practice, which has the characteristics of multi-type, multi-media, cross-time, dispersion, cross-language and disorder. The task of traditional information resource integration is basically limited to the structured category, and lacks targeted and purposeful investigation and processing. Whereas, there is a clear need in hazard management for typhoon events. It is necessary to learn from similar historical cases and quickly deduce the various scenarios and trends that may evolve from typhoon events according to the characteristic information of each stage, in order to provide support for hazard prediction and emergency decision-making. Knowledge service organizes, assembles, and refines a variety of explicit and implicit information resources that can be used, and then makes it orderly, thereby facilitating the use of knowledge to solve specific practical application problems (Colombo, 2011). Therefore, the transition from information services to knowledge services has become an inevitable requirement for hazard management.

Knowledge representation is the basis and premise of knowledge application. On the one hand, knowledge needs to be represented before organization and storage. On the other hand, knowledge representation determines the subsequent processing efficiency and application scope. Essentially, knowledge representation is the modeling, symbolization and formalization of knowledge. In the context of big data, knowledge representation is closely related with the strategy of transforming human knowledge into a system control structure that can be processed by computers. Typhoon event knowledge representation method is not the association and combination of all kinds of typhoon information products, but based on the typhoon evolution process and human cognition, through combing and summarizing the inherent mechanism of typhoon events, the formation of typhoon events can be characterized by various concepts and conceptual relations.

By using the knowledge of typhoon events represented by TKRM, various types of typhoon information can be re-integrated more completely. Abstract common elements of typhoon events from the knowledge level to reveal the evolution process and node status of each typhoon event. Furthermore, typhoon event knowledge based on the TKRM model can provide an accurate and effective knowledge environment for solving problems such as historical typhoon information management, activity mode mining, and typhoon forecasting.

The modifying content of revised manuscript details refer to the lines 39-57 on the 1st-2nd page and the lines 390-396 on the 16th page.

Point 2: Lines 13-16: Is this a decision support system for typhoons? Can we use this system for typhoon analysis and prediction or disaster management (Leung et al., 2012; Ding et al. 2015)?

Response 2: The aim of this study is to construct the knowledge representation method of typhoon events. The main results include: (1) typhoon event knowledge representation model; (2) formal method of typhoon event knowledge. Knowledge representation makes it possible for human knowledge to be transformed into a system control structure that computers can process. Based on the knowledge representation method, it can provide reference and basis for the transformation from various complicated typhoon information to typhoon events knowledge. At present, the sharp contrast between the sudden increase of mass information and the lack of knowledge appreciation mechanism appears in the age of big data. Using the TKRM can further promote the transformation from information service to knowledge service in disaster management field.

The modifying content of revised manuscript details refer to the lines 13-18 on the 1st page and the lines 101-106 on the 3rd page.

Point 3: Line 18: consists of

Response 3: We agree with the reviewer's comment that there is a misrepresentation. We have modified the contents of manuscript as follows:

… by analyzing the evolution characteristics of typhoon events, the TKRM framework with 3 layers consists of "event-process-state" is constructed, …

The modifying content of revised manuscript details refer to the lines 23 on the 1st page.

Point 4: Line 50: “are composed into” might not be the best option in this context.

Response 4: We agree with the reviewer's comment that there is a misrepresentation. We have modified the contents of manuscript as follows:

… the contents of different indicators describing typhoons are formed into time series according to the sequence of their generation time. …

The modifying content of revised manuscript details refer to the lines 65-66 on the 2nd page.

Point 5: Line 289: What is the trigger condition? How can we use this trigger condition?

Response 5: The TKRM formal method proposed is based on FSM, so the basic operation mechanism of TKRM formal method is the same as that of FSM. A state qm in FSM with given input σj will be transferred to a new state qn determined by δ. Of these, σj is the input character and is the external factor which causes the state change in FSM. The δ is a state transfer rule, which is used to determine whether σj meets the requirements of state transfer. FSM can realize the transfer between states only when σj fulfills the relevant rule of δ.

The TKRM formal method is an extension of FSM. Due to the increase in the modeling of spatio-temporal features, σj is split into 3 parts: (1) T∑={tσ0, tσ1, ..., tσn} is the set of time conditions, L∑={lσ0, lσ1, ..., lσn} is a set of location conditions, A∑={aσ0, aσ1, ..., aσn} is a set of input conditions other than time and location, and is also defined as the set of trigger conditions. Therefore, the trigger condition A∑ is also part of the input character and is used the same way as σj.

The modifying content of revised manuscript details refer to the lines 230-242 on the 8th page and the lines 264-270 on the 9th page.

Point 6: Lines 300-302: In this method similar to “decision tree” or “association rule”? For example, Yang et al. (2011) and Zhang et al. (2013) applied decision tree and association rule methods to examine tropical cyclones. Some conditions could lead to a decision on the state of tropical cyclones.

Response 6: Although decision tree and association rule can also express event structure or relationship between events, their application scenarios have a different emphasis. In the TKRM representation model, time and location are the basic frameworks for the existence of typhoon events, and each element of the event is organized according to spatiotemporal clues. Formal methods need to embody these characteristics all above.

In the case of association rules, it is mainly used to discover the correlation between different elements in an event. When the support and confidence of association rules meet the threshold given by users, the co-occurrence of antecedent and consequent can be judged. In fact, although the variations of different features (such as air pressure, wind speed, etc.) in typhoon events may be correlated, the typhoon states following the spatiotemporal framework do not co-occur. Typhoon events are always in continuous change, and the different states alternate with each other.

The decision tree is a tree structure, each node represents a decision, each decision can further lead to two or more decisions and lead to different results. The premise of constructing a decision tree is to extract the regular features that repeatedly appear in the event, and based on the known probability of occurrence in various situations. Obviously, there is indeed some regularity in the changes in the attributes and behaviors of typhoon events. However, regarding the most important spatiotemporal characteristics of event evolution, it is difficult to extract the regularity of change that satisfies any typhoon. Even if different typhoon events may have similar evolution trends, the spatial-temporal range of their existence is also different.

In comparison, FSM, as a mathematical model for describing a finite number of states and their behaviors such as transitions and actions, fits well with the structure of the TKRM representation model. Besides, FSM is suitable for describing sequential and logical things. In this study, FSM was chosen instead of decision tree and association rule mainly because FSM is better at reflecting and representing the dynamics of events.

The modifying content of revised manuscript details refer to the lines 243-251 on the 8th page.

Point 7: Line 348: There is lack of

Response 7: We agree with the reviewer's comment that there is a misrepresentation. We have modified the contents of manuscript as follows:

…there is lack of a unified knowledge description mechanism for typhoon events …

The modifying content of revised manuscript details refer to the lines 392 on the 16th page.

Point 8: Lines 394-397: The authors wrote “In addition, the combination of the TKRM-based knowledge service model with disaster prevention and mitigation, emergency decision-making and other fields will be explored to achieve the effect of supporting the knowledge guarantee for crisis early warning, monitoring, command, evaluation and other decision-making work.” Can the authors use some examples to demonstrate this aspect?

Response 8: The combination of the TKRM-based knowledge services with hazard prevention and mitigation, emergency decision-making and other fields will be explored. The details are as follows:

(1) Early warning. The evolution law of similar historical typhoon events should be fully based on. Comparing the characteristics of the current typhoon event with other typhoon events in history, the hazard severity and evolution trend of similar typhoon events can provide a reference for the current typhoon event prediction and early warning.

(2) Hazard monitoring, TKRM clearly defines the content scope of typhoon events. It should focus on acquiring and perceiving the content in time, location, attribute, behavior and influence.

(3) Prevention command, judging the state of typhoon event is the key to emergency response. According to the different states that the typhoon is in, guide and support decision-makers to accurately grasp the current situation. For example, in typhoon Lekima event, when the typhoon is in the state of "typhoon + upgrading", the meteorological departments need to closely monitor the movement direction and intensity change of the typhoon, and the potential impact areas need to be prepared in advance for emergency preparedness and hazard prevention. When the typhoon is in the "super typhoon + landfall" state, the affected areas need to start emergency plans, and hazard prevention and mitigation departments begin to implement hazard rescue. When the typhoon is in a "tropical storm + downgrading" state, the relevant departments and institutions need to carry out hazard relief and a series of rehabilitation and reconstruction work.

(4) Hazard assessment, based on the abundant content covered in TKRM, it can meet the needs of various assessments such as casualties, economic losses, and secondary disasters.

Through the above measures, the effect of supporting the knowledge guarantee for crisis early warning, monitoring, command, evaluation and other decision-making work is achieved.

The modifying content of revised manuscript details refer to the lines 461-476 on the 18th page.

Point 9: Line 409: The authors use “granularity” many times, but never giving an explanation. This should be properly defined for the readers to understand its meaning.

Response 9: Granularity is the basic unit of knowledge, and the size of granularity is a measure of the level of knowledge abstraction and knowledge included. According to the requirements of problem solving, knowledge is divided from different angles or levels, and each knowledge block is a knowledge granularity. For typhoon events, people need both integrated knowledge with larger granularity, such as the start and end time of the typhoon, the range of impact, and the overall hazard level, as well as detailed knowledge with smaller granularity, such as the landing location of typhoon, a typhoon's wind scale at a given moment, etc. .

The multi-granularity of typhoon event knowledge is affected by the multi-granularity of typhoon event, and it is also the need for different typhoon knowledge services. In this study, typhoon events are abstracted into 3 kinds of knowledge granularity of event, process and state. Events with larger granularity can be divided into processes and states with smaller granularity, and states with smaller granularity can be aggregated into processes and events with larger granularity. In TKRM, based on the hierarchical modeling method, an “event-process-state” typhoon event knowledge representation model is proposed.

The modifying content of revised manuscript details refer to the lines 148-162 on the 4th page.

Point 10: I can spot “Error! Reference source not found” throughout the manuscript. Please make sure this is corrected.

Response 10: We thank the reviewer for the valuable comment. The formatting of the entire manuscript has been revised according to the template.

Point 11: Moderate English changes required.

Response 11: We thank the reviewer for the valuable comment. We have checked and modified the English grammar and expression of the entire manuscript.

Reviewer 2 Report

Dear authors,

Thank you for you paper, nevertheless I have some remarks.

General ones:

You used many times the term „typhoon disasters“, but I should stress that we have a „typhoon hazard“ and if there will be interaction with the society with strong negative effects (!) than we can speak about disaster. It means if you describe the physical essence of the natural process it is in the frame of hazard (not yet disaster). To speak about the disaster, there must be already to society included. You have carefully check where you describe typhoon as natural process and where you already connected it with the society impact.

At many places you have „reference source not found“ e.g. line 101.

The chapter Methodology is missing completely (!). Why? This is not acceptable. You should explain for instance that you have theoretical part and case study included in one paper.

Specific comments:

Line 30: ….lead to „mountain torrent“, you would like probably to say ….to „flooding“. Because mountain torrents exit also without typhoon influence at all mountains around the word and only some of them cause problem during flooding events.

Line 31: the mud-rock flow is a wrong term, you probably mean „debris flow“, could be also „mud-flow“ – depends on the material involved in the flow.

Fig. 1: those internal relations are certainly important but I am not sure how relevant they are for the paper in such a general way. And you have here mistakes or at least questionable links. For instance: rainstorms can surely trigger floods, but not mentioned here. Strong wind can trigger landslides? I can imagine that it might be at some really special (!) occasional cases a co-influence, but not really a trigger. You can find much stronger links. In such a way is this Fig. only for limited use.

L124-125 (with relation to the general remarks): Typhoon is natural hazard and what it makes it a disaster is the vulnerability of the society, if the hazardous process occurs in highly populated area.

Table 1: in the column “Types” you mixed different types of storms with behaviour – in one column? Why? The headline is wrong in this case.

L 346: it is not true that typhoons are interaction of synoptic system, surface environment and the society in all cases. It could be, but typhoons exist also in non populated areas. I guess that it will be better to write atmosphere instead of synoptic system.

Discussion: I don not see to many references in this chapter and it should be primary about relation to other published works about this topic (apart of limits of use).

Author Response

Point 1: You used many times the term „typhoon disasters“, but I should stress that we have a „typhoon hazard“ and if there will be interaction with the society with strong negative effects (!) than we can speak about disaster. It means if you describe the physical essence of the natural process it is in the frame of hazard (not yet disaster). To speak about the disaster, there must be already to society included. You have carefully check where you describe typhoon as natural process and where you already connected it with the society impact.

Response 1: We thank the reviewer for the valuable comment. According to the comments, the meanings of hazard and disaster are distinguished and the usage of different words in the revised manuscript is clarified. Because this study is mainly aimed at the knowledge modeling of tropical cyclone evolution process based on event model, the meaning of typhoon hazard is expressed in most cases. Besides, there are also some contents about the influence of typhoon on human living environment and social activities in some cases. In such cases, use typhoon disaster to express it.

Point 2: At many places you have „reference source not found“ e.g. line 101.

Response 2: We thank the reviewer for the valuable comment. The formatting of the entire manuscript has been revised according to the template.

Point 3: The chapter Methodology is missing completely (!). Why? This is not acceptable. You should explain for instance that you have theoretical part and case study included in one paper.

Response 3: We agree with the reviewer that we should partially adjust the structure of the article. In fact, the TKRM Framework and TKRM formalization describe the knowledge representation method of typhoon events. The knowledge representation model is proposed in TKRM Framework and the formal method is proposed in TKRM formalization.

Therefore, the “Methodology” section consists of a combination of the TKRM framework and TKRM formalization. The original “case study” section, which verifies the validity and feasibility of TKRM, is adjusted to the “Experiment and results” section.

The modifying content of revised manuscript details refer to the 2nd–16th page.

Point 4: Line 30: ….lead to „mountain torrent“, you would like probably to say ….to „flooding“. Because mountain torrents exit also without typhoon influence at all mountains around the word and only some of them cause problem during flooding events.

Response 4: We agree with the reviewer's comment that there is a misrepresentation. We have modified the contents of manuscript as follows:

… but also easily lead to flooding, debris flow, landslide and other secondary disasters …

The modifying content of revised manuscript details refer to the lines 35-36 on the 1st page.

Point 5: Line 31: the mud-rock flow is a wrong term, you probably mean „debris flow“, could be also „mud-flow“ – depends on the material involved in the flow.

Response 5: We agree with the reviewer's comment that there is a misrepresentation. We have modified the contents of manuscript as follows:

… but also easily lead to flooding, debris flow, landslide and other secondary disasters …

The modifying content of revised manuscript details refer to the lines 35-36 on the 1st page.

Point 6: Fig. 1: those internal relations are certainly important but I am not sure how relevant they are for the paper in such a general way. And you have here mistakes or at least questionable links. For instance: rainstorms can surely trigger floods, but not mentioned here. Strong wind can trigger landslides? I can imagine that it might be at some really special (!) occasional cases a co-influence, but not really a trigger. You can find much stronger links. In such a way is this Fig. only for limited use.

Response 6: We thank the reviewer for the valuable comment. Because the expression of Fig. 1 is too complicated, and there are some uncertainties. Therefore, Fig. 1 is removed from the revised manuscript. At the same time, the description of typhoon evolution process has been added to reflect the typhoon characteristics at various stages of its life cycle.

The modifying content of revised manuscript details refer to the lines 128-135 on the 3rd-4th page.

Point 7: L124-125 (with relation to the general remarks): Typhoon is natural hazard and what it makes it a disaster is the vulnerability of the society, if the hazardous process occurs in highly populated area.

Response 7: We agree with the reviewer's comment on the inaccuracy of the words used in the original expression. We have modified the contents of manuscript as follows:

In essence, people's cognition of typhoon events is to abstract the complex typhoon weather system and its natural environment. If the occurrence of typhoon causes damage to human production and life, it will further cover the abstraction of social environment.

The modifying content of revised manuscript details refer to the lines 166-168 on the 5th page.

Point 8: Table 1: in the column “Types” you mixed different types of storms with behaviour – in one column? Why? The headline is wrong in this case.

Response 8: We agree with the reviewer's comment that there is a misrepresentation. We have modified the contents of manuscript as follows:

… classification …

The modifying content of revised manuscript details refer to the lines 260 on the 6th page.

Point 9: L 346: it is not true that typhoons are interaction of synoptic system, surface environment and the society in all cases. It could be, but typhoons exist also in non populated areas. I guess that it will be better to write atmosphere instead of synoptic system.

Response 9: We agree with the reviewer's comment that there is a misrepresentation. We have modified the contents of manuscript as follows:

…Typhoon events are usually the result of the interaction of atmosphere, surface environment and human society system …

The modifying content of revised manuscript details refer to the lines 390-391 on the 16th page.

Point 10: Discussion: I don not see to many references in this chapter and it should be primary about relation to other published works about this topic (apart of limits of use).

Response 10: We agree with the reviewer's comment that discussion should be primary about relation to other published works about this topic. The structure of the discussion part is reorganized and adjusted, and the research results are discussed from advantage, application and recommendations 3 perspectives.

In knowledge modeling, compared with static conceptual model and timeslice snapshot model, the TKRM can reflect the characteristics of typhoon evolution process better. By using the knowledge of typhoon events represented by TKRM, various types of typhoon information can be re-integrated more completely. Abstract common elements of typhoon events from the knowledge level to reveal the evolution process and node status of each typhoon event.

In application potential, typhoon event knowledge based on the TKRM model can provide an accurate and effective knowledge environment for solving problems such as spatial-temporal pattern, evolution law, activity mode and internal mechanism. Therefore, TKRM is not only the basis of general knowledge services such as retrieval, analysis and prediction, but also can provide targeted support for typhoon prevention and reduction.

In recommendations for future research, the combination of the TKRM-based knowledge service mode with hazard prevention and mitigation, emergency decision-making and other fields will be explored to achieve the effect of supporting the knowledge guarantee for crisis early warning, monitoring, command, evaluation and other decision-making work.

In each section of the discussion, the literature on the subject is cited, reflecting the relationship to published works about the topic.

The modifying content of revised manuscript details refer to the lines 397-476 on the 16th-18th page.

Round 2

Reviewer 1 Report

I appreciate the authors to revise this manuscript following my comments. The revision reads smooth. I suggest to accept it at the present form. 

Reviewer 2 Report

Dear authors,

thank you for the revised version and I already agree with publishing.

best regards, reviewer